# Development of Predictive Equations for Thermal Conductivity of Compost Bedding



**Flávio A. Damasceno [1,\*], Joseph L. Taraba [2], George B. Day [2], Randi A. Black [3], Jeffrey M. Bewley [4], Tales J. Fernandes [1], Carlos E. A. Oliveira [5], Rafaella R. Andrade [5], Matteo Barbari [6], Patrícia F. P. Ferraz [1] and Lorenzo Leso [6]**

[1] Department of Engineering, Federal University of Lavras, Lavras 37200-000, Brazil; tales.jfernandes@ufla.br (T.J.F.); patricia.ponciano@ufla.br (P.F.P.F.)
[2] Department of Agricultural Engineering, University of Kentucky, Lexington, KY 40502, USA; joseph.taraba@uky.edu (J.L.T.); george.day@uky.edu (G.B.D.)
[3] Division of Agriculture and Natural Resources, University of California, Davis, CA 95403, USA; rablack@ucanr.edu
[4] Holstein Association USA, Brattleboro, VT 05302, USA; jbewley@holstein.com
[5] Department of Agricultural Engineering, Federal University of Viçosa, Viçosa 36570-900, Brazil; carloseoliveira@ufv.br (C.E.A.O.); rafaella.andrade@ufv.br (R.R.A.)
[6] Department of Agriculture, Food, Environment and Forestry, University of Firenze, 50145 Firenze, Italy; matteo.barbari@unifi.it (M.B.); lorenzo.leso@unifi.it (L.L.)
\* Correspondence: flavio.damasceno@ufla.br; Tel.: +55-35-3829-4515

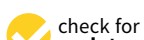



**Featured Application: The findings are highly relevant for the design and management of bedded pack barns.**

**Abstract:** Among animal facilities, compost-bedded pack (CBP) barns have attracted a lot of attention from milk producers and the scientific community. Systematic investigation of the main thermal properties utilizing sawdust in CBP barns is of environmental and economic relevance. In this paper, the aim was to (a) develop predictive equations for the thermal conductivity ($k$) of compost bedding as a function of moisture content (MC), the degree of compaction ($DC_o$), and particle size (PS); and (b) investigate the links between $k$ and depth within bedding material. Samples of compost bedding materials were collected from 42 commercial CBP barns distributed throughout Kentucky (USA). From these predictive equations, it was possible to understand how the MC, $DC_o$, and PS of the bedding materials may influence the behavior of $k$. These results are very useful for solving obstacles to simulate and predict the variable outcomes of the compost bedding materials process in CBP barns, allowing for its optimization, consequently reducing the time and energy spent on their optimization and allowing for simulation and assessment of compost bedding process modifications. The results of the current study may have important implications in the design and management of bedded pack barns.

**Keywords:** bedding material; dairy cow; modeling; prediction; thermal properties

## 1. Introduction

In the late 1980s, innovative dairy producers in the state of Virginia (USA) introduced a new loose-housing system for dairy cattle, generally referred to as a compost-bedded pack (CBP) barn [1]. Since then, CBP barns have been introduced in many other US states, and in other countries [2]. In CBP barns, cows are provided with an open bedded pack area where cows can stand, walk, and rest without restriction to areas or postures. Research showed that CBP barns have the potential to improve animal welfare, but results strictly depend on pack management, especially pack moisture content [2,3].

CBP barns require periodic bedding addition and a recommended twice daily stirring with some agricultural implement (roto-tiller or deep-tillage tool) to incorporate and mix

manure and air into the pack while exposing a greater pack surface area for drying [4–6]. This process promotes microbiological activity, heating and drying the pack and providing a fresh, dry surface for cattle to lie on [7].

Adequate temperature and moisture content in the pack must be maintained for effective composting [5]. The recommended internal temperature for CBP at depths of 15–31 cm ranges from 40.0 to 65.0 °C and optimum moisture content from 40 to 60% [7].

Depending on climate and composting activity, large quantities of bedding materials may be needed in CBP to absorb excessive moisture and maintain an optimum moisture level in the pack [5].

The most common bedding materials used in CBP are wood byproducts including sawdust, wood shavings, and wood chips [7,8]. In some countries, such as Brazil and Argentina, some milk producers have used coffee husks, peanut husks, sugar cane bagasse, and rice straw as bedding material in CBP [6,9,10]. Such materials, especially sawdust, can be either fresh or dried. Dried sawdust is preferred over fresh sawdust as the latter may contain high moisture, which reduces water absorption [11].

A CBP barn operating at optimal moisture levels should provide lower rates of ammonia conversion and minimize nitrous oxide production. Studies indicated that a high MC of >65% and low C/N ratio resulted in higher $N_2O$ emission rates [12,13], which typically result from using too little bedding, resulting in sub-optimal composting conditions. This system, coupled with its relatively low operating costs, will provide dairy farmers with a tool to reduce the emission of greenhouse gases to the atmosphere once correctly managed. This may, in turn, reduce the impact of this animal production system on climate change.

Compared to windrow composting, CBP barns have a larger surface area to heat generating volume, thus more heat losses. Maintaining a high pack temperature is important in CBP barns [14]. This requires an understanding of the thermal properties and heat balance in composting pack systems, which can potentially improve CBP management and design [4,15]. In recent years, considerable effort has gone into developing techniques to determine the thermal properties of composting materials as they are relevant in many areas of agriculture engineering, agronomy, and animal science [16,17].

Thermal properties of composting materials affect the temperature and biodegradation rate during the composting process. Well-determined thermal properties of compost feedstock will therefore contribute to practical thermodynamic approaches and mathematical models involving heat and mass transfer [18]. However, it can be very expensive and physically impractical to obtain many of the parameters. Therefore, feasible simplifications are to be sought, and various compromises are to be adopted in computational simulation case studies that emphasize either the model complexity or material data accuracy [19].

Thermal conductivity, specific heat capacity, and thermal diffusivity are the three most important thermal properties regarding heat transfer analysis [15], and these are used in engineering design calculations involving the thermal processing of agricultural products. In agricultural materials, temperature and moisture content greatly influence the thermal properties due to the relatively high specific heat, thermal conductivity, and thermal diffusivity of water [20]. These three thermal properties can be measured by several methods.

Although thermal properties are very important in composting, information of their values for various compost materials is lacking. In general, it is difficult to determine the thermal properties of moist materials because forced heating during the measurements causes internal liquid and/or gas convection. This often results in overestimating the thermal properties. The thermal probe method is the most attractive method because it uses relatively simple equipment to determine the thermal properties for moist materials [21].

Thermal conductivity (*k*) refers to the intrinsic ability of a material to transfer or conduct heat [22]. Therefore, the *k* of composting materials affects temperature through conduction and the rate of biodegradation during the composting process [15]. Studies have shown that *k* can assist in the monitoring of the moisture content of compost

material in different layers with the fundamental requirements for understanding the composting process [23].

The $k$ can assist in the monitoring of the moisture content of compost bedding material in different layers with the fundamental requirements in both animal welfare and pack management.

In this context, the determination of $k$ is extremely important for the calculation of heat and mass transfer between the bedding material and the animals. Accurate transfer models will enable: (1) the selection of the most suitable materials for the composting process, (2) with the dimensioning of ventilation systems that promote surface drying of the bedded pack, and (3) the ability to infer bed management requirements by predicting, for example, the amount of heat and moisture produced during the composting process with the depth of the pack. In the short term, these mathematical models could provide the research community with an approach to design systems with cleaner exhaust air and lower impacts on animals as well as those individuals that live or work in the surroundings of these agricultural facilities. Thus, the thermal properties of bedding materials are used for designing an ideal agricultural implement to manage the pack and improve the composting process.

The $k$ depends on several factors, such as texture, organic matter, water content, compaction degree, and bulk density [23]. Therefore, estimating $k$ values through these mathematical models can reduce the complexity for those designing CBP barns. These models circumvent the high costs of experiments to empirically quantify this parameter in the field.

The development of mathematical models and the application of computer simulations allow us to reduce the time and costs of development and renovation projects. The $k$ of composting material varies subject to several factors, including the content of organic compounds, density (specific weight), porosity [19,24], and moisture content [25]. Those parameters fluctuate in different phases of the composting process [26].

The objectives of the current study were to: (a) develop predictive equations for the thermal conductivity ($k$) of compost bedding as a function of moisture content (MC), the degree of compaction ($DC_o$), and particle size (PS); and (b) investigate the links between $k$ and depth within bedding material. These data are of interest in the domains of environmental pollution, biosystems engineers, bioresource technology, and, more generally, heat transfer in porous media.

## 2. Materials and Methods

### 2.1. Sample Collection

Samples of compost bedding materials were collected from 42 commercial CBP barns distributed throughout the state of Kentucky (USA). At each farm, samples of compost bedding were collected from the 0–10 cm surface layer in 9 evenly distributed locations throughout the resting area (Figure 1). Bedding samples were collected using an iron hoe and soil auger. A 20 L container was filled with incremental quantities of bedding collected from the 9 locations to obtain a composite sample of each CBP. The bedding samples were immediately refrigerated upon return to the lab, at 1.0 °C. Depending on the type of material used as bedding, the samples collected were classified as: (a) green sawdust (GS), (b) kiln-dried wood shavings or sawdust (KD), and (c) a mix of both (MX).

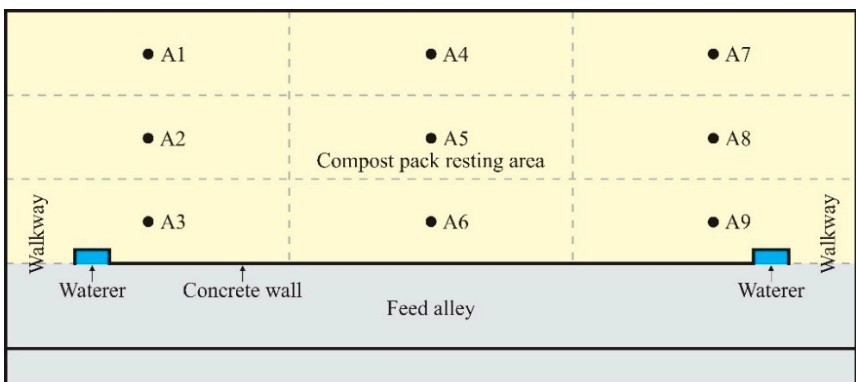

**Figure 1.** Nine grid spaces (A1 to A9) of sample collections inside the CBP barns.

## 2.2. Sample Preparation

Samples of compost material with 3 levels of MC (30%, 45%, and 60%) were produced and mixed in a concrete mixer for 3 min. The desired levels of MC were achieved by adding distilled water to the samples during mixing. The amount of distilled water to be added was calculated based on Maia et al. [27]. If the initial MC was higher than the target MC, the material was weighed and left to air-dry until reaching the target MC. The range of MC (30%, 45%, and 60%) was selected to simulate typical conditions found in CBP [2].

As-received bedding samples were allowed to air-dry for 48 h before determination of the PS distribution. Dried compost was poured in graduated volume cylinders sieved for 3 min in a sieve shaker (Ro-Tap Model B, W. S. Tyler, Inc., Mentor, OH, USA) with sieves vertically aligned in series in a decreasing mesh screen opening order: 25.00 mm, 8.00 mm, 5.60 mm, 4.75 mm, 2.00 mm, and a pan of the bottom. The amount of compost retained in each screen was poured in a beaker and its weight determined. Details to determine PS distribution can be seen in Maia et al. [28].

## 2.3. Simulation of Material Compaction

To allow the simulation of different DCo, a mechanical pressing device was designed and constructed at the shop of the Biosystems and Agricultural Engineering at Department of the University of Kentucky (Lexington, KY, USA). The press device (Figure 2) was assembled and located in the Agricultural Air Quality Laboratory of the same department. A fluffy bedding material was weighed, and the PVC cylinder was filled. The PVC cylinder was then vibrated for 60 s using vibrating jigsaw (Black+Decker, Model JS515, Towson, MD, USA), and, if necessary, more bedding sample was added to the PVC cylinder until the total volume was completed.

In this study, PVC was used as the material for the PVC cylinder (sample vessel) because: (a) it is easy to handle and clean; (b) it can withstand pressure and long-term loading; and (c) it is resistant to corrosion and relatively light and cost-effective. In addition, the effects of wall friction in the sample vessel needed to be minimized during compression for the applied load to be translated into compression of the sample. Thus, the internal walls of the PVC cylinder have low friction [29].

The sample volume must be large enough to accommodate a representative sample of bedding material. Therefore, a PVC cylinder of approximately 4.7 L was designed, with a diameter of 0.15 m and a height of approximately 0.27 m. This PVC cylinder had a series of three equally spaced holes through the entire height. A detailed drawing of the PVC cylinder is shown in Figure 2.

The target $DC_o$ was achieved by a mechanical pressing device that was developed to pack the sample down in the PVC container (Figure 2). The compaction level inside of the PVC cylinder was adjusted using a pneumatic lever valve and analog pressure gauge that controlled the air pressure inside of the steel cylinder chamber that moved the steel piston. A moisture drain valve prevented water condensation in the line of air pressure. The

pressures applied (0.0, 0.1, 0.2, 0.3, and 0.4 MPa) were based on studies conducted by Van der Tol et al. [30,31], which evaluated the pressure distribution under the bovine claw. The application of dynamic pressure occurred with the opening and closing of the pneumatic lever valve 12 times per minute, causing the vertical displacement of the compression valve to change. This amount was based on observations of the number of average steps per minute performed by dairy cows. A schematic drawing of the overall system is presented in Figure 3.

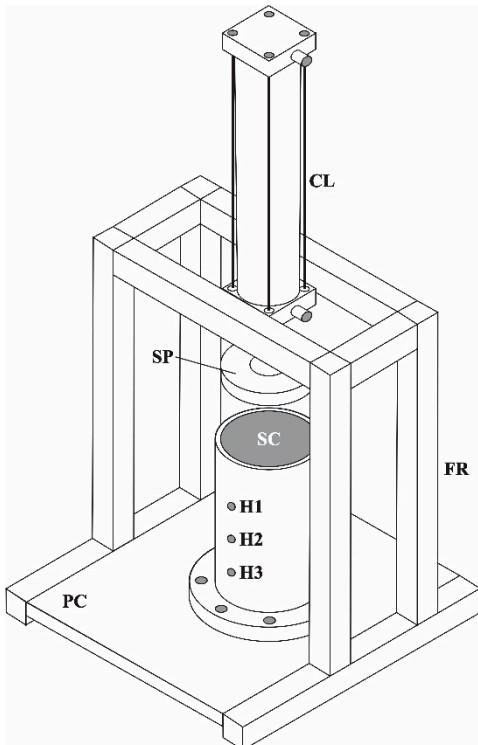

**Figure 2.** Designed compost compact device used in this study. Legend: PC—base and platen cover; SC—PVC sample cylinder; FR—steel frame; CL—cylinder; SP—steel piston; and H1, H2, and H3—holes over cylinder used to measure the thermal properties.

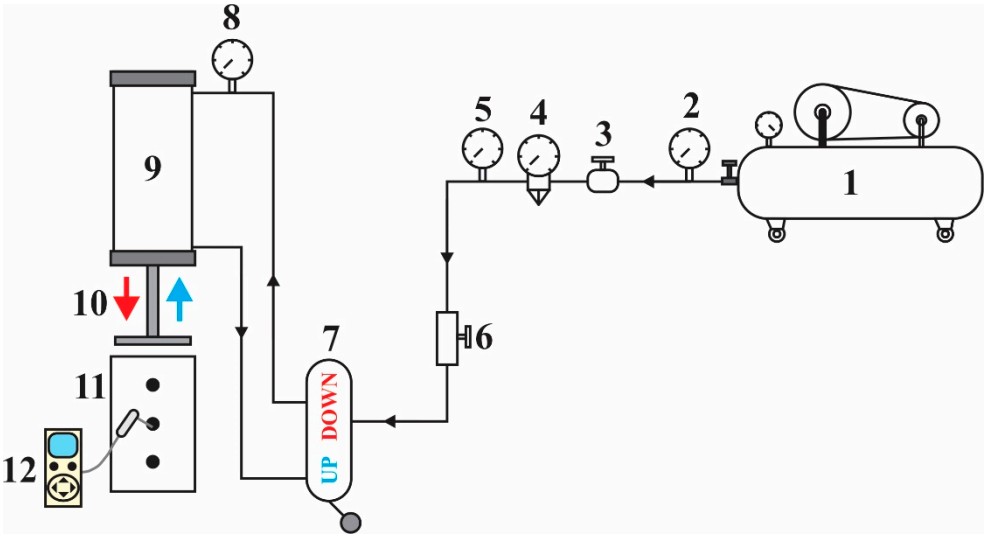

**Figure 3.** Schematic drawing of overall system: 1—Air compressor; 2—Analog pressure gauge; 3—Pressure relief valve; 4—Moisture drain valve; 5—Analog pressure gauge; 6—Safety valve; 7—P-

neumatic lever valve (Up and Down); 8—Analog pressure gauge; 9—Steel cylinder chamber; 10—Steel piston; 11—PVC cylinder (sample vessel); and 12—Handheld transient heat dissipation device.

Dynamic pressure was expected to produce a different *k* compared with static pressure. All of the tests were carried out in two iterations: with static compaction (SC) and with dynamic compaction (DC).

### 2.4. Thermal Conductivity Measurement

The experiment was conducted under ambient room temperatures. This temperature environment is in the thermal range where the heat generated by potential microbial respiration would be negligible; <10% of peak respiration rate at optimum MC range [32]. The thermal conductivity (*k*) of composting materials was determined at all possible combinations of PS levels (0.00 mm < Finer < 2.00 mm; 2.00 mm < PS2 < 4.75 mm; 4.75 mm < PS3 < 5.60 mm; 5.60 mm < PS4 < 8.00 mm; 8.00 mm < Coarser > 8.00 mm), MC levels (30; 45; and 60%—w.b.), and $DC_o$ levels (0.0; 0.1; 0.2; 0.3; and 0.4 MPa).

A handheld transient heat dissipation device (KD2, Decagon, Pullman, WA, USA) was used to determine *k* of samples (Figure 3). The sensor was calibrated by factory and performance verification standards were observed during this study. Measurements were made by placing the sensor probe into the three holes in the PVC cylinder wall at different depths (H1 = 7.5 cm, H2 = 15.0 cm, and H3 = 22.5 cm) and recording the measurement after one minute (Decagon Devices, 2016).

### 2.5. Statistical Analysis

To truly quantify the several parameters' impacts on the pack compost process, a quantitative assessment of each of its components must be performed, and the relevant physical processes must be included in this assessment. One aspect that has been typically neglected is the impact that material compaction has on the coupled water and heat fluxes across the pack layers. In this work, we developed some mathematical models that used empirical bedding material data to consider how compaction levels and moisture changed the substrate's thermal conductivity.

A randomized experimental design with two replicates was analyzed in two different analyses. The first statistical analysis with each experimental unit consisted of three MC levels (30%, 45%, and 60%) and five $DC_o$ levels (0.0; 0.1; 0.2; 0.3; and 0.4 Mpa) with each dairy farm assessed as a repetition (42 farms). Therefore, each bedding sample contained 630 observations.

Prior to analysis, the original dataset that consisted of 630 observations was split into a training set (70% of the data) and a test set (30% of the data). Variable selection and model fitting were performed on the training set while the test set was used for model validation.

Statistical analysis was performed using the software R version 3.4.1 [33]. Analysis of variance (ANOVA) was performed considering the factorial scheme with two factors, to assess the possible dependence between the factors MC and $DC_o$ in the description of the response variable *k*. This ANOVA was performed using the *lm* function available on the basis of the statistical software R (R Core Team, 2019). The ANOVA was performed to assess the effects of MC and $DC_o$ on the *k* of bedding materials and the possible interaction between the two factors (MC $\times$ $DC_o$).

The ANOVA was performed, using the package *stats* [33], to assess the effects of MC and $DC_o$ on the *k* of bedding materials. The fixed effects of MC and $DC_o$, as well as their interaction (MC $\times$ $DC_o$), were included in the models. After this analysis, based on the significance of the statistical tests for MC, $DC_o$, and their interactions, the appropriate regression models were chosen to describe *k*. The selection of models was made, using the package *stats* [33], based on backward stepwise procedure starting from a complete model with all the predictive variables to the quadratic term. Non-significant predictors were removed from the model based on the relative reduction in sums of squares.

In all analyses, a polynomial model was adjusted taking into account the interaction between the two explanatory variables (MC and $DC_o$), generating a response surface [34].

The selection of models was made based on backward stepwise procedure starting from a complete model with all the predictive variables to the quadratic term. Non-significant predictors were removed from the model based on the relative reduction in sums of squares. In view of this, a polynomial model was fitted, taking into account the dependent variables and the interaction between these variables, as described in Equation (1).

$$k_i = b_0 + b_1 \cdot \text{MC} + b_2 \cdot DC_o + b_3 \cdot SC^2 + b_4 \cdot \text{MC} \cdot DC_o \tag{1}$$

where $b_0$, $b_1$, $b_2$, $b_3$, and $b_4$ are the parameters of the polynomial regression model. The curve intercept is $b_0$ and $b_1$, $b_2$, $b_3$, and $b_4$ represent the degree of influence of the respective variable on the thermal conductivity ($k_i$). The $b_1$, $b_2$, $b_3$, and $b_4$ are the linear coefficients, that is, keeping the others constant and adding a unit of measure in the explanatory variable associated with one of these coefficients, an increase or decrease in the response is expected.

In order to obtain better precision in the modeling of $k$, the analyses were evaluated for the three depths separately (H1, H2, and H3). Separate analyses were also carried out for SC and DC.

In the second statistical analysis, the mean values of $k$ for each PS level (Finer, PS2, PS3, PS4, and coarser), three depths (H1, H2, and H3), three MC levels (30%, 45%, and 60%), and five $DC_o$ levels (0.0; 0.1; 0.2; 0.3; and 0.4 MPa) with two replicates from 42 bedding samples, totalizing 18,900 observations, were analyzed by ANOVA (package *stats*; R Development Core Team, 2019). In this case, the statistical analyses were performed separately because they are different variables. The parameters of the models were estimated using the least squares method also with the *lm* function of software R (R CORE TEAM, 2019). The polynomial regression and Student's *t*-test were used. ANOVA (package *stats*; R Development Core Team, 2019) was used to evaluate and compare measured versus simulated $k$. The $R^2$ values of the polynomial regression indicated how consistently the measured versus predicted values follow a best-fit line, ranging from 0 (no correlation) to 1.0 (perfect correlation).

The response surface (based on parameter estimates) and data graphs were plotted using Sigma Plot software version 12.0 (Systat Software Inc., San Jose, CA, USA). The differences between the group means (MC, SC, DC, compost particle range, and depth) were tested by the Tukey test (Figures 4, 6, 9–11). For all analyses, significance level was set at $p \leq 0.05$.

## 3. Results and Discussion

### 3.1. Thermal Properties of Composts in Different Moisture Contents and Static Compaction Degrees

The values of $k$, recorded during the study, are presented graphically as a function of each percentage (30, 45, and 60%) of the MC evaluated and as a function of the different SCs (0.0, 0.1, 0.2, 0.3, and 0.4 MPa) of the bedding materials in Figures 4a and 4b, respectively. These results indicate that the values of $k$ increased as MC and SC increased, likely due to a reduction in the volumetric fraction of air within the bedding material [35]. The $DC_0$ was not statistically significant according to the model selection method (backward stepwise), and depths (H1, H2, and H3) were evaluated separately.

The greatest variation in $k$ values for MC and SC was $\pm$ 0.061 W m$^{-1}$ K$^{-1}$ and $\pm$ 0.060 W m$^{-1}$ K$^{-1}$, respectively, where they define desirable levels of accuracy by the thermal sensor to estimate the thermal conductivity. Considering the physical mechanism associated with conduction in general, the $k$ of a solid > liquid > gas [36].

Due to the large number of outliers (Figure 4), the discrepancy of the $k$ values, which are applied in a modeling study, can reduce the accuracy of the model results, as already observed in other studies [15,19,37]. However, it can be highlighted that even with the large number of outliers, the mathematical models adjusted in this study presented good quality of fit, as can be observed in the $R^2$ values (>91%). This behavior can be explained by the large amount of data (640 values) that was used to fit mathematical models.

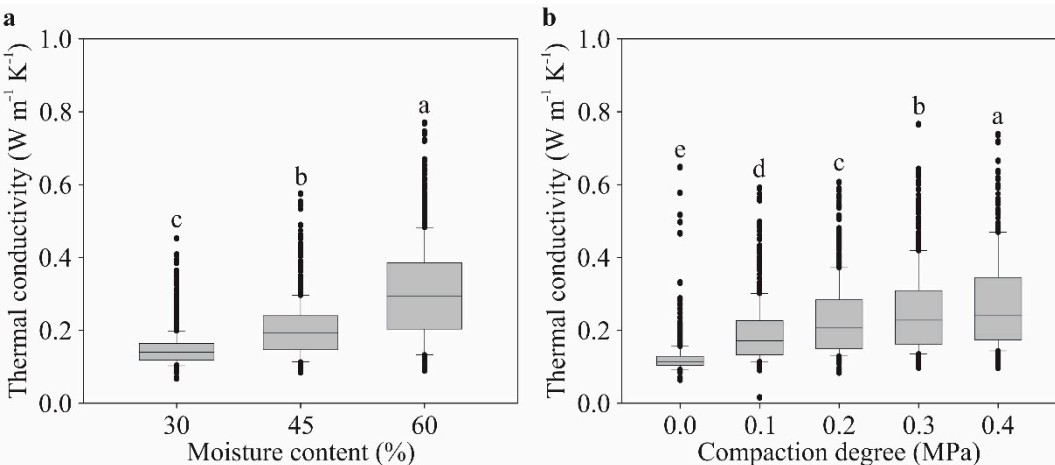

**Figure 4.** Variation in thermal conductivity (*k*) at different degrees of moisture content (**a**) and (**b**) static compaction degree of compost bedding materials. Values followed by different letters are significantly different ($p < 0.05$; Tukey).

The analysis of variance showed that for the three depths, there was a significant interaction between SC and MC, regarding its effect on *k* ($p < 0.001$). Thus, it is verified that the levels of SC and MC are not independent to explain the values of *k*. Then, after descriptive analysis of the data, through the dispersion graph, it was found that the behavior of the variable MC (30%, 45%, and 60%) is of a linear line, while the behavior of the variable SC (0.0, 0.1, 0.2, 0.3, and 0.4 MPa) is a quadratic model.

The values found for *k*, in the different compost bedding materials, in relation to the levels of MC (30, 45, and 60%) and SC (0.0, 0.1, 0.2, 0.3, and 0.4 MPa) tested for each one of the three depths evaluated (H1, H2, and H3) ranged from 0.0141 to 0.768 W m$^{-1}$ K$^{-1}$.

The regression analysis for compost materials showed a polynomial relationship between the variable of *k* and the dependent variables MC and SC for each depth evaluated in this way:

$$k_{H1} = 0.0022 \cdot MC + 0.1838 \cdot SC - 1.1601 \cdot SC^2 + 0.0154 \cdot MC \cdot SC, \quad R^2 = 0.92 \tag{2}$$

$$k_{H2} = 0.0028 \cdot MC + 0.6711 \cdot SC^2 + 0.0140 \cdot MC \cdot SC, \quad R^2 = 0.92 \tag{3}$$

$$k_{H3} = 0.0027 \cdot MC + 0.4251 \cdot SC^2 + 0.0111 \cdot MC \cdot SC, \quad R^2 = 0.91 \tag{4}$$

A significant effect was observed when observing the MC and SC separately and also when observing the significant influence of the MC × SC interaction, since methodologically it is already known that the *k* value of water is greater than that of air (0.06 W m$^{-1}$ K$^{-1}$ > 0.026 W m$^{-1}$ K$^{-1}$; Bergman et al. [22]). The thermal conductivity of air is about 25 times less than that of water; hence, if air is replaced by water in the pores of a material (i.e., if the water content increases), the bulk thermal conductivity increases [38].

Therefore, increasing the moisture content, the pores of the bed material were filled with water, which increased the values of *k*. In addition, *k* also increased with the increased degree of SC due to a reduction in empty space. Generally, the thermal properties of composting materials show specific tendencies related to water content, apparent density, and particle size [15,35,39,40]. If water content increases, the volume fraction of air decreases, and the volume fraction of solid increases. Consequently, the bulk thermal conductivity increases [38]. In other words, if SC and MC increase, solid particles move closer to one another, and the thermal contact resistance between bed particles decreases, thus raising the *k* values.

The highest values of *k* were observed at depth H1 (0.153 ± 0.047 W m$^{-1}$ K$^{-1}$), followed by H2 (0.142 ± 0.045 W m$^{-1}$ K$^{-1}$) and H3 (0.136 ± 0.041 W m$^{-1}$ K$^{-1}$), respectively. Values of *k* in several other studies have varied depending on the type of material, com-

posting time, moisture content, and physical and chemical characteristics, among other factors. Iwabuchi and Kamide [41] reported that the *k* of the dry compost (dairy manure and sawdust mix) was 0.051 W m$^{-1}$ K$^{-1}$ and 0.096 W m$^{-1}$ K$^{-1}$ with 57% moisture. As noted by Iwabuchi et al. [21], the values of *k* of composting material (dairy manure and sawdust mix), with moisture content ranging from 0 to 44.2%, showed values between 0.05 and 0.202 W m$^{-1}$ K$^{-1}$, respectively.

The response surfaces presented in Figure 5 were quadratic due to the adjusted polynomial regression models, showing good correlation between the adjusted data for the different depths evaluated ($R^2 > 91\%$), and the parameters of the models were significant ($p < 0.05$).

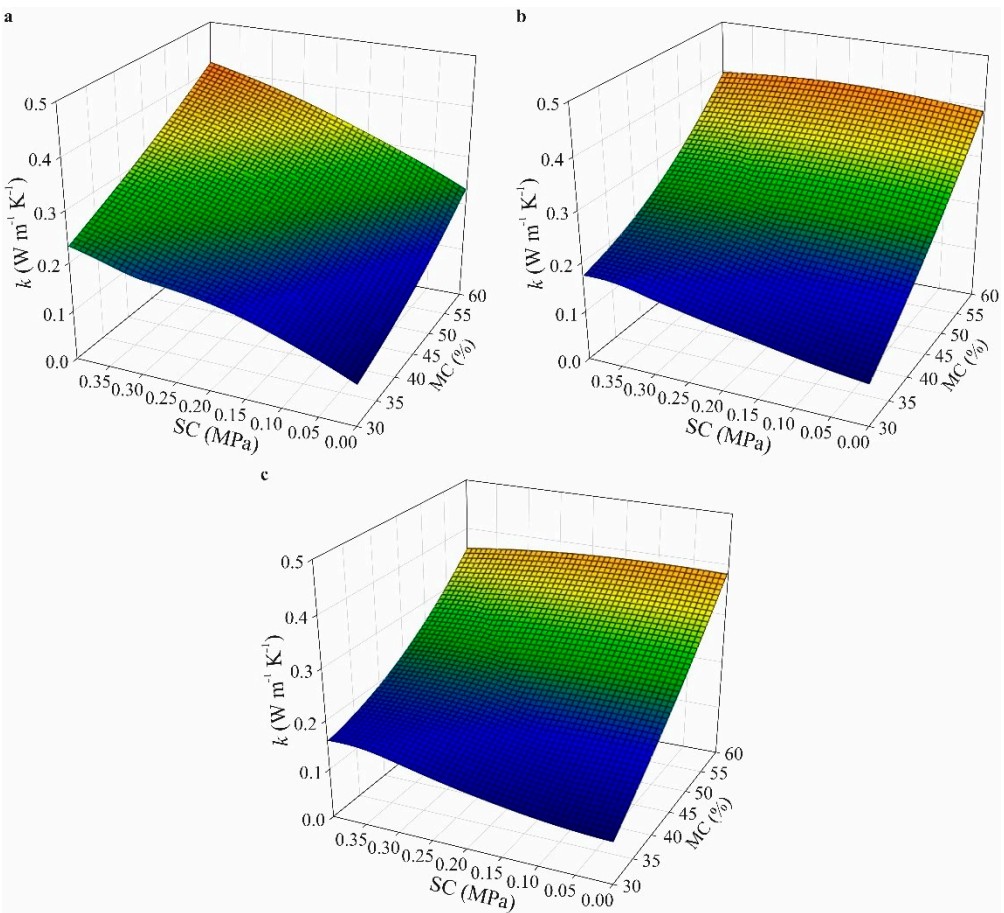

**Figure 5.** Response surface of moisture content (MC) and static compaction (SC) on thermal conductivity (*k*) of bed compost materials at different depths: H1 (**a**), H2 (**b**), and H3 (**c**).

For all compost materials tested (Figure 5), *k* values increased as MC and SC increased. The composting material with low moisture content presented low thermal conductivity due to the presence of a relatively high fraction of air, thereby reducing heat propagation with uniform temperature distribution across layers of bedding material. Since water, air, and solid materials have their own specific thermal property values, the thermal properties of compost materials vary with the proportion of these three materials, changing the water content, apparent density, and particles [42–45]. Usually, materials that have a low density also have a low thermal conductivity; consequently, these materials are poor for heat storage. In other words, the thermal conductivity of a bedding material is generally characterized by its dependence on particle density, as the void space is reduced, the filler particles begin to contact each other, and a continuous path is formed through the volume of the sample, increasing the heat and mass transfer with the environment. This fact can be

observed in the field, as when stirring the pack in a CBP barn, the fast exchange of heat and mass is observed between the bedding material and the environment.

In Figure 5, it is also possible to observe that the behavior of $k$ as a function of SC is slightly different in depth H1 in relation to the other depths tested (H2 and H3). This difference was shown by the polynomial model when presenting in Equation (2) a linear term (0.1838·SC) that was not significant in Equations (3) and (4).

According to Zuñiga et al. (2009), the thermal conductivity of porous media shows a greater significant difference between the degrees of compaction and depths tested. This could be due to the fact that heat conductivity is controlled by all three phases (solid/liquid/gas) of the material, thus influencing the $k$ values along the evaluated depth. In addition, the properties of a porous medium itself such as material texture, the insulating properties of any organic layers, and material compaction enhancing the decrease in porosity may endogenously affect material thermal properties [46,47].

### 3.2. Thermal Properties of Composts in Different Moisture Contents and Dynamic Compaction Degrees

As can be observed in Figure 6, $k$ presented higher values with a 60% MC and a DC of 0.4 MPa (0.789 W m$^{-1}$ K$^{-1}$). Theoretically, these values of $k$ can be predicted, since the DC over time has a direct effect due to the increase in the apparent density of the bedding material, reducing the porosity and resulting in an increase in $k$ values. The thermal conductivity of leaf compost increased linearly with the increase in water content and compaction degree, represented by volume fractions of air [35].

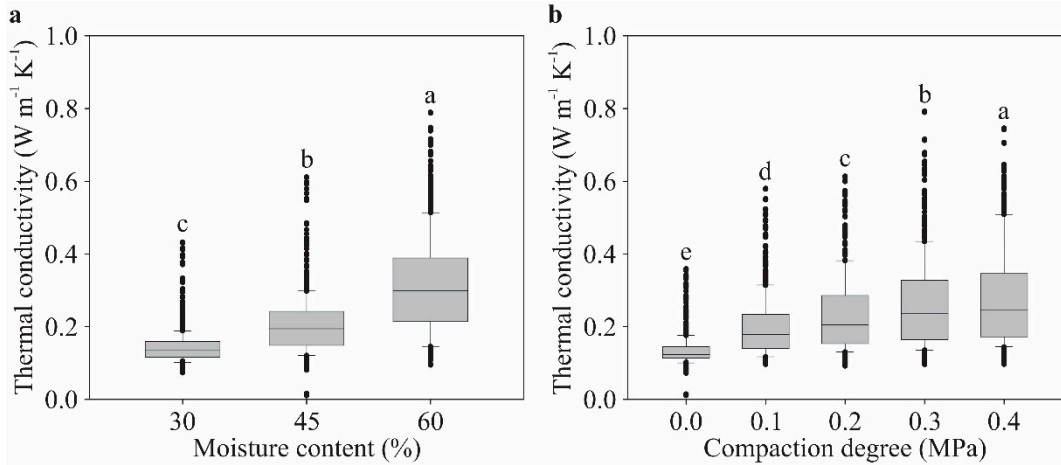

**Figure 6.** Variation in thermal conductivity ($k$) at different degrees of moisture content (**a,b**) dynamic compaction degree of compost bedding materials. Values followed by different letters are significantly different ($p < 0.05$; Tukey).

The ANOVA indicated that there is a significant correlation ($p < 0.05$) between the values of DC and MC along the three depths evaluated (H1, H2, and H3). Thus, in the same way as with SC, there is a dependence of DC and MC on the values of $k$, and a polynomial regression model was adjusted taking into account the interaction between DC and MC for each evaluated depth (H1, H2, and H3).

The multiple regression analyses for compost materials showed a polynomial relationship between $k$ and the dependent variables MC and DC for each depth evaluated (H1, H2, and H3), and the selected models were:

$$k_{H1} = 0.0029 \cdot MC - 0.7641 \cdot DC_2 + 0.0147 \cdot MC \cdot DC, \; R^2 = 0.91 \tag{5}$$

$$k_{H2} = 0.0030 \cdot MC - 0.7583 \cdot DC_2 + 0.0146 \cdot MC \cdot DC, \; R^2 = 0.92 \tag{6}$$

$$k_{H3} = 0.0029 \cdot MC - 0.4251 \cdot DC_2 + 0.0137 \cdot MC \cdot DC, \; R^2 = 0.91 \tag{7}$$

The magnitudes of the respective determination coefficients (R2) in Equations (5)–(7) confirm the good quality of fit obtained by the polynomial models in the description of $k$ by the variables DC and MC. Thus, the models are appropriately adjusted based on the observation of high value F as well as a high coefficient of determination (R2).

As shown in Figure 7, there was an increasing trend in the values of $k$ in the composting material with increasing MC and DC. The values of $k$ with different MC contents and degrees of DC varied from 0.010 to 0.789 W m$^{-1}$ K$^{-1}$. In this case, the standard deviation of $k$ was $\pm$ 0.015 W m$^{-1}$ K$^{-1}$, which is an acceptable precision to estimate the $k$ when using a transient line heat source sensor (Decagon KD2-Pro, accuracy $\pm$ 0.02 W m$^{-1}$ K$^{-1}$).

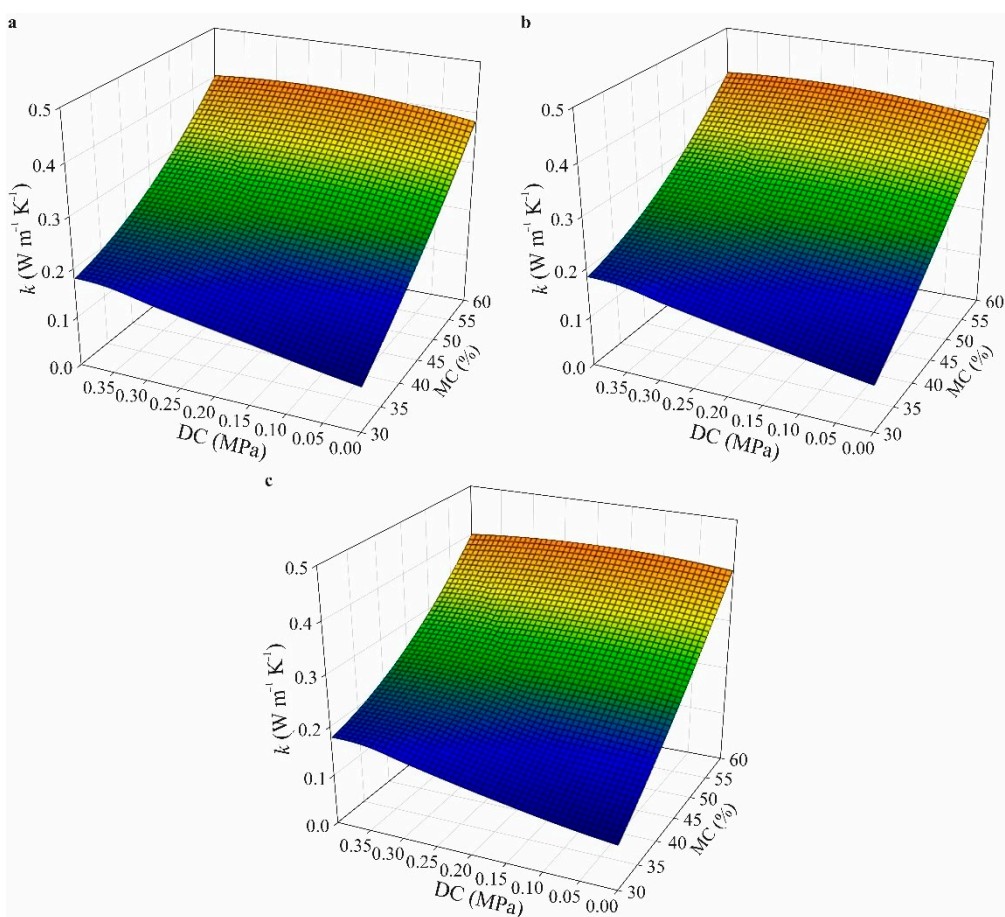

**Figure 7.** Response surface of moisture content (MC) and dynamic compaction (DC) on thermal conductivity ($k$) of bed compost materials at different depths: H1 (**a**), H2 (**b**), and H3 (**c**).

In Figure 7, contrary to what was observed in the behavior of $k$ as a function of SC (Figure 5), the influence of DC on the values of $k$ did not vary along the depths tested. Thus, Equations (5)–(7) have the same components (MC and DC), with only the estimates of the parameters varying ($k_{H1}$, $k_{H2}$, and $k_{H3}$). This probably occurred because of the pressure variation applied by the plunger in the bed samples, which may have rearranged the particles of the materials, reducing the porosity and density over the entire depth tested. This behavior is consistent with the findings of other investigations [48] and occurs because as the material is dynamically compacted, there is more physical contact between the solid particles, which increase thermal conduction [49]. Additionally, the magnitude of the thermal conductivity is similar to that presented in the literature [48–50].

*3.3. Thermal Properties of Composts in Different Particle Sizes*

The $k$ of granular materials is known to be affected by the quality of contact between the particles. The particle contact quality depends on the material density, particle geometry,

and PS distribution. The effects of these factors on *k* are evaluated in this study using air-dried bedding materials and transient line heat source sensors.

The behavior of directly observed *k* data in relation to PS distribution for bedding samples in compost barn plants using green sawdust, greenhouse sawdust, or a mixture of materials is shown in Figure 8. Measurements of *k* were not carried out on samples of thicker material whose particle size was larger than 25 mm due to the difficulty in inserting the sensor thermal probe into the sample.

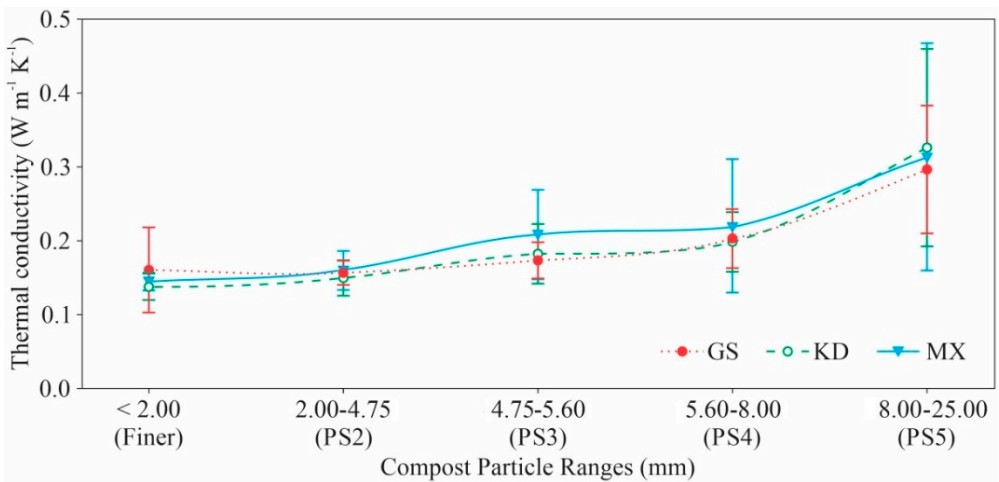

**Figure 8.** Variation in thermal conductivity (W m$^{-1}$ K$^{-1}$) with different compost particle sizes of compost materials: Green sawdust, Kiln-dried shavings or sawdust, and Mix.

As can be observed in Figure 8, the values of *k* increase with the increase in the PS of the analyzed materials, which is the reverse of what was expected, as described by Yun and Santamarina [51] and Ahn et al. [15]. According to Singh et al. [36], the porosity has a great influence on the thermal properties of the material due to its dependence on the PS, that is, as the PS increases, the thermal conductivity decreases, since more particles are required for the same porosity, meaning more thermal resistance between the particles. This behavior can probably be explained by the higher retention of moisture in larger particles since the compost bedding materials were air dried for the same period of time and because the materials had different PSs, so drying did not occur evenly, which probably influenced the values of the thermal properties. Considering the physical mechanism associated with conduction in general, the *k* of a solid is greater than that of a liquid and that of a liquid is greater than that of a gas [36]. Thus, the effect of PS on the *k* was more pronounced on larger PSs. However, a small increase in *k* for all materials was observed when PS was less than 4.75 mm. It was also observed that *k* increased dramatically (>85%) in materials with a PS greater than 8.00 mm (Figure 8). These changes can be explained considering the change in porosity filled with fine particles of material. For grain-like materials, for example, where the pores of the materials are filled with air, generally the *k* increases as the PS decreases. However, in materials with large particles, the *k* presents inverse results [16].

Statistical analysis of the data revealed that there was a significant difference between the values of *k* and the different PSs evaluated (Tukey test, $p < 0.05$). Thus, in Figure 9, the main effect of PS on *k* is presented. In general, *k* values were higher in compost materials with a PS greater than 8.00 mm.

This means that with a high PS (>8.0 mm), the impact of the MC on *k* became positive, in other words, depending on the MC, the *k* tends to increase when the PS increases. An explanation for this phenomenon may be that the increased PS created more space between the pores of the material, increasing the amount of water retention and raising the values of k. According to Cosenza et al. [25], as with other thermal properties, the *k* of the material depends on the MC, but the porosity and *k* of the solid fraction are also strong determinants.

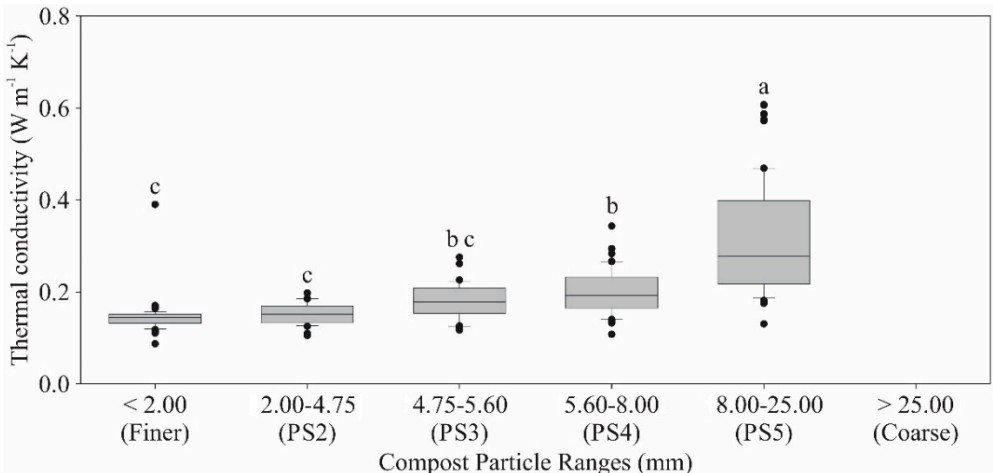

**Figure 9.** Variation in thermal conductivity (W m$^{-1}$ K$^{-1}$) and compost particle ranges (mm) for bedding materials. Values followed by different letters are significantly different (*p* < 0.05; Tukey test).

The average *k* of the bedding materials tested in the current study was 0.198 W m$^{-1}$ K$^{-1}$, varying between 0.088 and 0.608 W m$^{-1}$ K$^{-1}$ depending on particle size. Values of *k* found in this study agree with some previous studies. Research performed by Ahn et al. [15] showed that the *k* ranged from 0.03 to 0.05 W m$^{-1}$ K$^{-1}$ for dry sawdust and 0.03 to 0.06 W m$^{-1}$ K$^{-1}$ for dry wood chips.

### 3.4. Thermal Properties of Composts along the Depths Evaluated

The behavior of the *k* as a function of the DC$_o$ (SC and DC) along the depths evaluated (H1, H2, and H3) is observed in Figures 10 and 11.

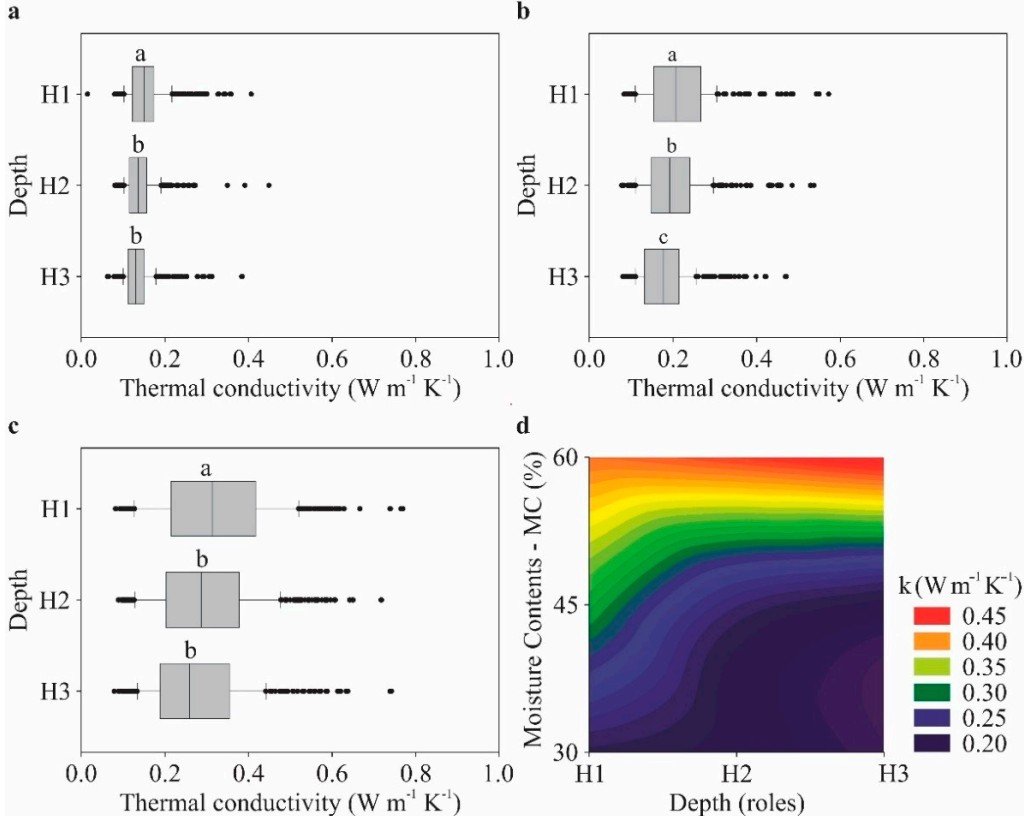

**Figure 10.** Thermal conductivity (*k*, W m$^{-1}$ K$^{-1}$) as a function of static compaction (SC) and depth (H1, H2, and H3) at different moisture contents: (**a**) 30%, (**b**) 45%, (**c**) 60%, and (**d**) interaction between

average values of SC and MC at different depths. Values followed by different letters are significantly different ($p < 0.05$; Tukey).

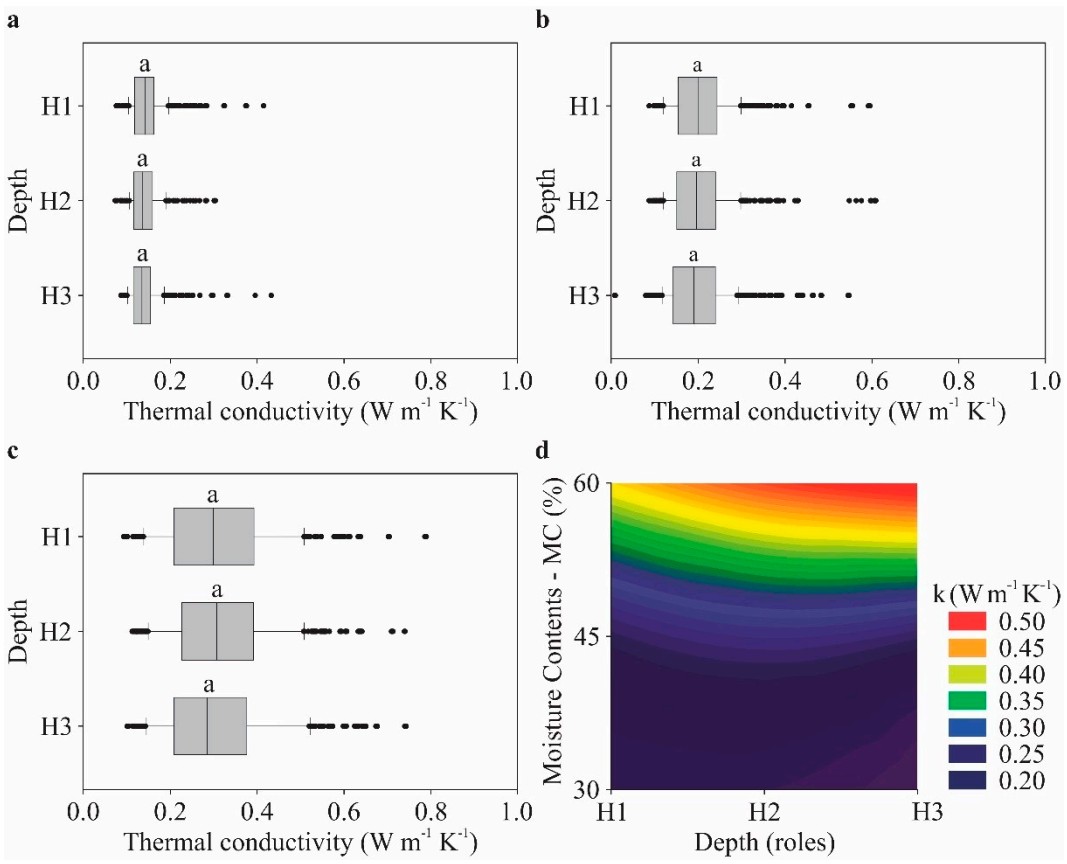

**Figure 11.** Thermal conductivity ($k$, W m$^{-1}$ K$^{-1}$) as a function of dynamic compaction (DC) and depth (H1, H2, and H3) at different moisture contents: (**a**) 30%, (**b**) 45%, (**c**) 60%, and (**d**) interaction between average values of DC and MC at different depths. Values followed by different letters are significantly different ($p < 0.05$; Tukey).

The values of $k$ increased with SC due to a reduction in void space and an increase in bulk density (Figure 10). However, statistical analysis revealed that depth alone did not have a significant impact or interaction with another factor. In general, the values of $k$ reduced along the evaluated depths, presenting higher values at depth H1. This upward trend was contrary to the study by Huet et al. [52], where it was found that depth alone did not have a significant impact or interaction with other evaluated factors (MC and PS) in thermal conductivity values.

However, when applying DC to the bedding material (Figure 11), it is observed that there is no significant difference in the values of $k$ along the depths evaluated (Tukey test, $p > 0.05$). The variation in DC may have influenced the bulk density of the material along any PVC cylinder. According to Lam et al. [53], the apparent density has a significant effect on the handling and storage of materials and depends on the composition of the material, PS, MC, and applied pressure, among others. The apparent biomass density increases during transportation, handling, and storage, which may be caused by compaction due to normal vibration, buoyancy, or loading [54].

### 3.5. Prediction of Bed Compost Thermal Conductivity

Figure 12 shows a comparison of the predicted $k$ from Equations (2)–(4) and the experimental data when applying SC for each depth evaluated (H1, H2, and H3).

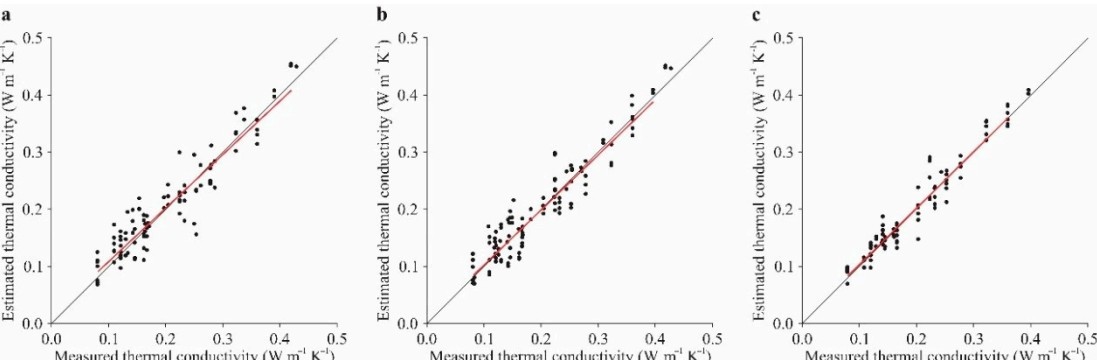

**Figure 12.** Estimated thermal conductivity ($k_{est}$, W m$^{-1}$ K$^{-1}$) of bed compost materials versus measured ($k_{mea}$, W m$^{-1}$ K$^{-1}$) values when applying static compaction (SC) for each depth evaluated: (**a**) H1, (**b**) H2, and (**c**) H3.

Through a preliminary statistical analysis, it was found that there is no statistically significant difference between the estimated thermal conductivity ($k_{est}$) and ($k_{mea}$) measured (*t* test, $p > 0.05$). The adjusted expressions are statistically significant (F test, $p < 0.0001$), providing mean errors of 0.065, 0.077, and 0.077, respectively (Figure 12).

When comparing the $k$ of bed compost materials applying SC values $k_{est}$ with the $k_{mea}$ values by Equations (2)–(4), the proposed models were found to be precise in predicting the $k$ (Figure 12). The adjustment equations between the models and the data for each depth (H1, H2, and H3) were:

$$k_{est\ H1} = 0.9444 \cdot k_{mea\ H1} + 0.0199, R^2 = 0.87 \tag{8}$$

$$k_{est\ H2} = 0.9604 \cdot k_{mea\ H2} + 0.0072, R^2 = 0.89 \tag{9}$$

$$k_{est\ H3} = 0.9936 \cdot k_{mea\ H2} + 0.0032, R^2 = 0.94 \tag{10}$$

According to Savegnago et al. [55], the coefficient of determination ($R^2$) is an indicator of the goodness-of-fit between the model and the data. Comparing $R^2$ values of the three test models showed that the third model (Figure 12c) outperformed the others (0.879, 0.895, and 0.940, respectively). These results indicate that the $k$ values predicted by the models (Equations (2)–(4)) were similar to values experimentally observed (Equations (8)–(10)) [56].

Figure 13 shows a comparison of the predicted thermal conductivities ($k_{est}$, W m$^{-1}$ K$^{-1}$) from Equations (5)–(7) and the experimental data ($k_{mea}$, W m$^{-1}$ K$^{-1}$) when applying dynamic pressure (DC) for each depth evaluated (H1, H2, and H3). The adjustment equations between the models and the data for each depth (H1, H2, and H3) were:

$$k_{est\ H1} = 0.8715 \cdot k_{mea\ H1} + 0.0101, R^2 = 0.79 \tag{11}$$

$$k_{est\ H2} = 0.9684 \cdot k_{mea\ H2} + 0.0013, R^2 = 0.83 \tag{12}$$

$$k_{est\ H3} = 0.9475 \cdot k_{mea\ H3} + 0.0072, R^2 = 0.86 \tag{13}$$

Based on Figure 13a–c, it is possible to observe that the estimated thermal conductivity of bed compost materials when applying DC for each depth evaluated in comparison with the observed values showed good results too. The adjusted expressions are statistically significant (F test, $p < 0.0001$), showing an $R^2$ of 0.794, 0.831, and 0.867, respectively, where the linear and angular coefficients are significant (*t* test, $p > 0.05$). The average and standard deviations of the simulated and observed $k$ at the three depths tested were $0.210 \pm 0.08$, $0.214 \pm 0.09$, and $0.209 \pm 0.09$, respectively, resulting in an average absolute deviation of 0.072, 0.078, and 0.074, respectively, and an average error of 0.037, 0.037, and 0.036, respectively. The model to predict the $k$ for H3 showed better results in comparison with H1 and H2.

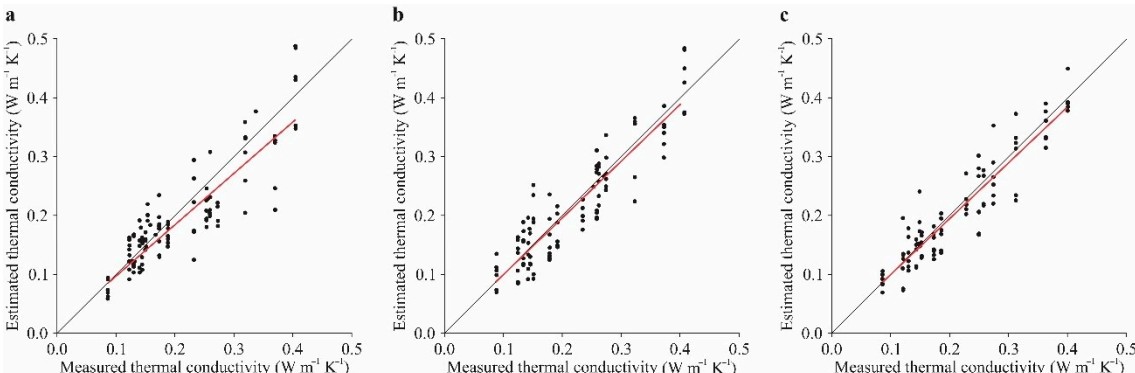

**Figure 13.** Estimated thermal conductivity ($k_{est}$, W m$^{-1}$ K$^{-1}$) of bed compost materials versus measured ($k_{mea}$, W m$^{-1}$ K$^{-1}$) values when applying dynamic compaction (DC) for each depth evaluated: (**a**) H1, (**b**) H2, and (**c**) H3.

In spite of the good R$^2$ values presented by Figure 13, these results indicate that the models to predict $k$ values applying SC (Figure 12) showed better adjustment with better R$^2$ values than the models to predict $k$ applying DC (Figure 13). Overall, the models proposed in this study to predict $k$ based on MC and DC$_o$ (both SC and DC) showed good generalization performance. The results are promising, considering that the modeled equations depend on variables that are subject to large variations in the field, such as moisture, applied pressure, the type of bed material, and depth.

## 4. Conclusions

Studies were conducted to evaluate the effect on thermal conductivity ($k$) with the variation in moisture content (MC), the degree of compaction (DC$_o$), particle size (PS), and the depth of bedding material from different compost-bedded pack (CBP) barns. Based on the results, the following conclusions are made:

(a) A strong dependence of thermal conductivity on the moisture content and degree of compaction (static and dynamic) was observed. Thus, a growing trend in thermal conductivity ($k$) was observed with increasing moisture content and degree of compaction for all bed materials tested. The polynomial regression models developed in this study presented an excellent model fit, with R$^2$ greater than 91%.
(b) Overall, thermal conductivity increased with increasing particle size (PS), indicating that it is strongly dependent on PS, and;
(c) In general, the values of thermal conductivity reduced along the evaluated depths.

**Author Contributions:** Conceptualization: F.A.D., J.L.T., G.B.D. and R.A.B.; data acquisition: F.A.D.; data analysis: F.A.D. and T.J.F.; design of methodology: F.A.D.; writing and editing: F.A.D., C.E.A.O., R.R.A., P.F.P.F., R.A.B., J.M.B., L.L. and M.B. All authors have read and agreed to the published version of the manuscript.

**Funding:** This research received external funding of the Brazilian National Research Council—CNPq (n. 407052/2018-6) and FAPEMIG (APQ-00853-17).

**Institutional Review Board Statement:** Not applicable.

**Informed Consent Statement:** Not applicable.

**Data Availability Statement:** https://www.locus.ufv.br/handle/123456789/726.

**Acknowledgments:** The authors would like to thank the Brazilian Organizations (CNPq, FAPEMIG, and CAPES) and the University of Kentucky, whose support was greatly appreciated.

**Conflicts of Interest:** The authors declare no conflict of interest.

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
