# Peer review of "Development of Predictive Equations for Thermal Conductivity of Compost Bedding"

_applsci, doi:10.3390/app11188503_

Round 1
Reviewer 1 Report
The work is of scientific interest and is well done. The introduction and conclusions can be improved. The introduction can be improved with more bibliography and the conclusions seem to short.
Author Response
Dear editor,
All considerations requested by reviewer 1 were accepted, however we do not agree to add further revisions to the introduction, as we believe that it is already too extensive and covers all the points of the objective of this paper. There was a significant reduction in the conclusion.
Thanks

Reviewer 2 Report
Esteemed Authors,
It has been a great honor, as well as a pleasantly challenging activity, to review the article entitled ”Development of predictive equations for thermal conductivity of compost bedding.”
Compost-bedded pack barns (CBP) are receiving increasing attention as a housing system for dairy cows that can improve animal welfare. In CBP, cows are provided with an open bedded pack area rather than the individual stalls and concrete alleys found in free-stall systems. The bedded pack, a mixture of organic bedding and cattle excreta, is cultivated frequently (1–3 times per day) to incorporate fresh manure and air into the pack, thus promoting an aerobic composting process.
To function well, CBP generally requires a large area per cow. Optimal animal densities over the bedded area range from 7.4 to more than 15 m2/cow depending on several factors, including climate, bedding, pack management, and cow characteristics. Most studies have indicated that CBP, compared with conventional systems such as free-stall barns, has the potential to improve the welfare of dairy cows. In particular, the main reported benefits include improved comfort during resting, better foot and leg health, and more natural animal behavior. Research has also indicated that adequate udder health can be achieved in CBP. However, because the bedded pack has been shown to contain high bacterial concentrations, proper management is essential to maintain adequate cow cleanliness and reduce the risk of mastitis. Controlling pack moisture is consistently indicated as a critical issue with CBP. Especially under cold and humid weather conditions, large amounts of bedding may be necessary to keep the pack adequately dry and comfortable for the cows. Nevertheless, the improvements in cow health may offset the higher costs of bedding.
This system used predominantly in dairy farms has many advantages, including economic and ecological. It allows the efficient use of many by-products from agriculture or timber exploitation, being environmentally friendly and helping to reduce pollution.
The physical properties of the bedding are fundamental both in terms of hygiene and microbiology. The thermal conductivity combined with a certain water content can have unfavorable effects on the development of some microorganisms and the survival of some species of parasites.
At the farm level, developing countries are characterized by specific features such as a small scale and a household subsistence character of producers. Therefore particular models need to be prepared to capture in low-income economies and at the farm level, and from there at a regional or national level, the impacts of different policy options.
Public policies for family farms are essential for areas where such farms are numerous: most of the time, in such regions, we witness a rather high degree of population poverty, which requires the intervention of the competent state authorities, through various mechanisms, to balance the economic and social situation.
Innovation and the use of new technologies provide win-win solutions in agriculture. This means combining benefits for the environment and climate while increasing efficiency and competitiveness. However, the uptake of new technologies remains far below expectations and varies from one country to another. This gap needs to be addressed to ensure that everyone, including the small and medium-sized farmers, can access technology and benefit from it.
Theory-wise, the paper is likely to elicit specialists' interest in areas such as farmers' behavior, economy, sustainable development of agriculture, sustainable development of the environment, pollution reduction, and increasing the productivity of agricultural resources, especially by-products. The paper presents essential practical applicability related to the environment, sustainable development of agriculture, farmers' education, animal welfare, and social awareness of farmers' management in animal production. Moreover, the obtained results are also relevant to the production sector, particularly certain agri-food products in developing countries.
The paper is well structured and possesses a strong novelty character. The article's major components – Introduction; Materials and Methods; Results and Discussion and Conclusions - are organized judiciously and directly linked to one another.
The documentation is adequate, and the provided scientific results are exact and precise. The goal of the conducted research is well specified and delineated. The working protocol is appropriate, and the used analysis methods are coherent with the proposed objectives.
The bibliography of the paper is generous. What is even more relevant for the article's quality, all the authors in the bibliographic reference list are quoted in the text of the material – with one exception – Campbell and Norman, 2012 - number 50 in the list of bibliographical references - which does not appear quoted in the text of the article.
The work also benefits from adequate iconographic support, materialized by one table and thirteen figures.
The obtained results are interpreted correctly, and their practical value is visible.
The graphical representation of the results is adequate; as for the grammar of the paper, most of the text is very well written, with very few parts that would require some minor corrections, as follows:
Page 6, line 213 – replace “All tests” with “All of the tests”;
Page 6, line 219 – replace “were determined” with “was determined”.
The paper also contains some minor mistakes related to the technical editing process, especially in the case of the list of bibliographic references. Minor corrections and clarifications notwithstanding, the authors’ work and obtained results are highly commendable. They bring significant added value to the paper and may constitute a launching pad for further valuable studies.
Provided that the authors verify the paper and perform the required corrections, the article may be accepted and published in Applied Sciences.
Best Regards,
Reviewer
Author Response
Dear editor,
All corrections have been made.
